# A Prospective, Randomized, Placebo-Controlled Study Assessing the Efficacy of Chinese Herbal Medicine (Huangqi Guizhi Wuwu Decoction) in the Treatment of Albumin-Bound Paclitaxel-Induced Peripheral Neuropathy

**DOI:** 10.3390/jcm12020505

**Published:** 2023-01-07

**Authors:** Yue Chai, Fang Zhao, Peizhi Ye, Fei Ma, Jiayu Wang, Pin Zhang, Qing Li, Jiani Wang, Wenna Wang, Qiao Li, Binghe Xu

**Affiliations:** 1Department of Medical Oncology, National Cancer Center/National Clinical Research Center for Cancer/Cancer Hospital, Chinese Academy of Medical Sciences and Peking Union Medical College, Beijing 100021, China; 2Nursing Department of the Cancer Hospital, National Cancer Center/National Clinical Research Center for Cancer/Cancer Hospital, Chinese Academy of Medical Sciences and Peking Union Medical College, Beijing 100021, China; 3Chinese Medicine Department of the Cancer Hospital, National Cancer Center/National Clinical Research Center for Cancer/Cancer Hospital, Chinese Academy of Medical Sciences and Peking Union Medical College, Beijing 100021, China

**Keywords:** breast cancer, Chinese herbal medicine, peripheral neuropathy, Huangqi Guizhi Wuwu decoction, clinical trial

## Abstract

Objective: This study aimed to evaluate the efficacy and safety of Huangqi Guizhi Wuwu decoction (HGWD), which is composed of five crude drugs (Astragali Radix, Cinnamomi Ramulus, Paeoniae Radix Alba, Zingiberis Rhizoma Recens, and Jujubae Fructus), in the treatment of albumin-bound paclitaxel (nab-PTX)-induced peripheral neuropathy (PN) in Chinese patients with breast cancer (BC). Methods: This trial was conducted at the National Cancer Center in China from January 2020 to June 2022. The eligible participants were assigned randomly in a 1:1 ratio to an HGWD group or a control group. The outcome measure was EORTC QLQ-CIPN20 questionnaire. Results: 92 patients diagnosed with BC were enrolled and randomized to either HGWD group (*n* = 46) or control group (*n* = 46). There were no significant differences in baseline characteristics between the two groups (*p* > 0.05). A statistical analysis of the sensory and motor functions of the EORTC QLQ-CIPN20 scores showed that patients in the HGWD group reported a larger decrease in CIPN sensory scores than those in the control group (*p* < 0.001). The EORTC QLQ-CIPN20 autonomic scores showed no statistical significance between the two groups (*p* > 0.05). Conclusions: HGWD packs could significantly improve patients’ nab-PTX-induced PN, increase the tolerance for nab-PTX-containing chemotherapy, and further improve the quality of life of patients with BC.

## 1. Introduction

Breast cancer (BC) has now surpassed lung cancer as the most prevalent type of cancer in women [1]. In 2020, there were an estimated 2.3 million new breast cancer cases in women (11.7%), followed by lung cancer (11.4%), colorectal cancer (10.0%), prostate cancer (7.3%) and stomach cancer (5.6%) [1]. Paclitaxel (PTX) is a commonly used cell-cycle phase-specific chemotherapeutic agent, the main role of which is regulating mitosis in M phase. Albumin-bound paclitaxel (nab-PTX) is a newly developed nanoparticulate formulation of paclitaxel binding to human serum albumin. PTX could result in severe peripheral neuropathy (PN) in up to 50% of patients with cancer [2]. PN is also a common adverse reaction to nab-PTX [3,4,5]. The prevalence and severity of nab-PTX-induced PN (around 80%) were higher than those of solvent-based PTX-induced PN (around 60%) [6,7]. The mechanism related to nab-PTX-related PN is the damage of dorsal root ganglia, microtubules, mitochondria and nerve endings caused by nab-PTX [8]. Nab-PTX-related PN is primarily characterized as peripheral sensory neuropathy, occasionally presented as motor and autonomic neuropathies. PN-related symptoms are manifest usually as pain, tingling, and numbness in the hands and/or feet [9]. Nab-PTX-related PN mainly occurred 24–72 h after the administration of nab-PTX. PN seriously affects the quality of life of patients after chemotherapy and also limits the application of chemotherapy drugs [10].

The current treatments for nab-PTX-induced PN mainly included cryotherapy and compression therapy [10], oral B group vitamin [11], and duloxetine. However, the treatment effect of these above methods is limited. Huangqi Guizhi Wuwu decoction (HGWD) is an herbal formula recorded in “Synopsis of the Golden Chamber” for improving limb pain, tingling, and numbness, which is composed of five crude drugs (Astragali Radix, Cinnamomi Ramulus, Paeoniae Radix Alba, Zingiberis Rhizoma Recens, and Jujubae Fructus) [12].

Recently, HGWD has been shown to be effective in the treatment of oxaliplatin- and diabetic-related PNs [13,14],but there are no prospective studies to explore the efficacy of HGWD in the treatment of nab-PTX-induced PN in patients with BC. Therefore, we conducted this prospective randomized controlled study to investigate the efficacy and safety of HGWD to prevent nab-PTX-induced PN in patients with BC. The primary aim of this study was to assess the prevention of nab-PTX-induced PN by immersing and washing limbs with HGWD packs when compared to the placebo.

## 2. Materials and Methods

### 2.1. Study Design

This open-label, single-center, prospective, randomized controlled trial was planned to be conducted in the Cancer Hospital, Chinese Academy of Medical Sciences/ National Cancer Center in China from January 2020 to June 2022. The eligible participants were assigned randomly in a 1:1 ratio to a study group (HGWD group) or a control group. PASS software (version 15.0) (Kaysville, UT, USA) was used to calculate the sample size. Assuming α = 0.05 and 1-β = 0.8, a total of 80 patients were needed for the aim of decreasing the grade 2−3 PN from 60% to 30%. Assuming a lost-to-follow-up rate of 15%, a total of 92 patients were needed.

Patients in the HGWD group were treated with HGWD packs + placebo, whereas patients in the control group were treated with placebo. The efficacy of HGWD packs was evaluated using EORTC QLQ-CIPN20 questionnaires. Nab-PTX-induced neurotoxicity of all participants was evaluated before nab-PTX-containing chemotherapy administration and every month after nab-PTX-containing chemotherapy until 3 months (4 weeks, 8 weeks and 12 weeks).

This study’s registration number is NCT05566457 (www.clinicaltrials.gov, accessed on 4 October 2022) and was approved by the Ethics Committee of the National Cancer Center in China (19/327-2111). Written informed consents were obtained from all participants before treatment.

### 2.2. Patients

The main inclusion criteria comprised: (1) histologically confirmed untreated BC; (2) female; (3) age from 18 to 65 years old; (4) Eastern Cooperative Oncology Group (ECOG) performance status (PS) score of 0 or 1; (5) at least receiving 2 cycles of nab-PTX (260 mg/m2); (6) grade 2 or greater nab-PTX-induced PN evaluated by Common terminology criteria for adverse events (NCI-CTCAE) (version 5.0); (7) estimated survival time > 6 months. The exclusion criteria were: (1) a history of diabetes or neurological disorders; (2) mental disorder; (3) abnormal hepatic functions (total bilirubin upper limit of normal (ULN), alanine transaminase (ALT)/aspartate transaminase (AST) ≥ 2.5 × ULN), renal functions (creatinine ≥ 1.5 × ULN), and hematological functions (absolute neutrophil count ≤ 1.5 × 10^9^/L, platelet count ≤ 80 × 10^9^/L, hemoglobin < 90 g/L); (4) peripheral vascular insufficiency; (5) failure to complete chemotherapy due to treatment-related adverse events (TRAEs); (6) a family history of a genetic neuropathy; (7) had received previous treatment with neurotoxic chemotherapy, including oxaliplatin, cisplatin, vinca alkaloid, etc.; (8) a history of allergy to Chinese medicine.

### 2.3. Treatment

The eligible patients were randomly assigned to 2 groups. Patients in the HGWD group (*n* = 46) immersed and washed limbs with HGWD packs, followed by smearing limbs with vitamin E and vitamin B12 for 14 days after receiving nab-PTX-containing chemotherapy. Patients in the control group only used vitamin E milk and vitamin B12 to smear the limbs three times per day for 14 days after receiving nab-PTX-containing chemotherapy. The composition of an HGWD pack was: 60 g Radix Astragali (Huangqi), 15 g Ramulus Cinnamomi (Guizhi), 15 g Paeonia lactiflora (Baishao), 15 g Gentiana (Qinjiao), 6 g Scorpio (Quan-Xie), 20 g Rhizoma Zingiberis Recens (Shengjiang), 20 Jujubes (Dazao), 30 g Geranium wilfordii (Laoguancao), 12 g radix sileris (Fangfeng), 30 g Spatholobus suberectus (Jixueteng), 15 g Ligusticum (Chuanxiong), 15 g Poria (Fuling), and 15 g Radixcyathulae (Chuanniuxi). Add 1500 mL water to boil the HGWD pack for 20 min twice a day, and pour the drug-containing solution into the basin. The drug-containing solution was maintained at 39 to 40 ℃ for immersing and washing limbs for 20 min twice a day for consecutive 14 days. HGWD packs were provided by traditional medicine department of Cancer Hospital Chinese Academy of Medical Science (Beijing, China). HWGD packs were all used at home. Patients were followed every month till 3 months.

### 2.4. Assessment

The outcome measure was the European Organization for the Research and Treatment of Cancer-Chemotherapy-induced peripheral neuropathy 20 (EORTC QLQ-CIPN20) questionnaire [15,16] (Appendix A), which consisted of 20 items that graded the degree of functioning affected by sensory (9 items), motor (8 items) and autonomic CIPN symptoms (3 items). Each item was scored from 1 to 4 (corresponding to answers of none, mild, moderate, and severe) on a Likert scale and subsequently the scores were summed up (total score range, 1–44). Sensory raw scale scores ranged from 1–36, motor raw scale scores ranged from 1–32, and autonomic raw scale scores ranged from 1–12 for males and 1–8 for females. Higher scores represented more symptoms and a worse quality of life. The previous study showed that the assessment of CIPN symptoms is preferable with patient-reported outcomes over clinician-reported outcomes. Thus, the EORTC QLQ-CIPN20 questionnaire was regarded as one of the most appropriate outcome measures [17]. The patients received EORTC QLQ-CIPN20 questionnaires before nab-PTX-containing chemotherapy administration, 4 weeks, 8 weeks, and 12 weeks after nab-PTX-containing chemotherapy. Toxicity was monitored and recorded every week during the chemotherapy period, and every month during follow-up. The NCI-CTCAE (version 5.0) was used to grade adverse events (AEs), based on symptoms and laboratory tests of patients.

### 2.5. Statistical Analysis

The statistical analyses were performed in the intention-to-treat (ITT) population, which consisted of all patients randomly assigned to treatment. Baseline patient characteristics and the incidence of TRAEs were compared using the Chi-squared test between the HGWD group and the control group. Changes in EORTC QLQ-CIPN20 scores before and after treatment were assessed with Wilcoxon matched-pairs rank tests. Mann–Whitney U tests were used to compare EORTC QLQ-CIPN20 scores between the HGWD group and the control group. A *p* value < 0.05 was considered statistically significant. Statistical analysis was conducted using SPSS 22.0 for Windows software (IBM Corp., Armonk, NY, USA).

## 3. Results

### 3.1. Baseline Patient Characteristics

A total of 92 female patients diagnosed with BC at Cancer Hospital, Chinese Academy of Medical Sciences/National Cancer Center in China from January 2020 to June 2022 were enrolled and then randomized to either the HGWD group (*n* = 46) or the control group (*n* = 46). All enrolled patients completed the treatment. The flowchart of study design and patient enrollment is shown in Figure 1. The basic characteristics of the two groups are shown in Table 1. Chi-square tests showed that there were no significant differences in baseline characteristics between the two groups (*p* > 0.05). The median age was 50 years (range, 23 to 65) in the HGWD group and 52 years (range, 29 to 65) in the control group, respectively. In the HGWD group, 17.4% of patients (*n* = 8) were stage Ⅰ, 17.4% of patients (*n* = 8) were stage Ⅱ, 50.0% of patients (*n* = 23) were stage Ⅲ and 15.2% of patients (*n* = 7) were stage Ⅳ. In the control group, 13.0% of patients (*n* = 6) were stage Ⅰ, 21.7% of patients (*n* = 10) were stage Ⅱ, 41.3% of patients (*n* = 19) were stage Ⅲ and 23.9% of patients (*n* = 11) were stage Ⅳ. The most common subtypes were Luminal B (34.8% in the HGWD group and 39.1% in the control group) and triple-negative BC (TNBC) (54.3% in the HGWD group and 43.5% in the control group). The median number of cycles of nab-PTX-containing chemotherapy in both groups was 6 (range, 4–6). The chemotherapy was administered every 14 days or 21 days via an intravenous drip. Thirty-seven patients underwent 21-day nab-PTX-containing chemotherapy regimens (4 patients received administration on days 1 and 8 every 3 weeks, and 33 received administration on day 1 every 3 weeks), and 9 patients underwent 14-day regimens (administration on day 1 every 2 weeks) in the HGWD group. Forty-three patients underwent 21-day nab-PTX-containing chemotherapy regimens (administration on day 1 every 3 weeks) and 3 patients underwent 14-day regimens (administration on day 1 every 2 weeks) in the control group. There was no significant difference in the number of cycles of nab-PTX-containing chemotherapy between the two groups (*p* = 0.063).

### 3.2. Outcomes

For the EORTC QLQ-CIPN20 assessment, the questionnaire completion rate was 100%. In the HGWD group, 45 patients (97.8%) showed a decrease in CIPN sensory score, 40 patients (87.0%) showed a decrease in CIPN20 motor score and 3 patients (6.5%) showed a decrease in autonomic CIPN score. In the control group, 28 patients (60.9%) showed a decrease in CIPN sensory score, 10 patients (21.7%) showed a decrease in CIPN20 motor score and 4 patients (8.7%) showed a decrease in autonomic CIPN score. No patient developed an improvement in CIPN score in the two groups after 12 weeks follow up.

As shown in Table 2, there were significant differences in the sensory and motor score between the HGWD and control group (4 weeks: sensory *p* < 0.001, motor *p* < 0.001; 12 weeks: sensory *p* < 0.001, motor *p* < 0.001). Mann–Whitney U tests were also conducted for the EORTC QLQ-CIPN20 autonomic raw scale scores with no statistical significance observed (4 weeks: autonomic CIPN *p* = 0.633; 12 weeks: autonomic CIPN *p* = 0.702).

A statistical analysis of the sensory and motor functions of the EORTC QLQ-CIPN20 scores showed that patients in the HGWD group reported a larger decrease in CIPN sensory score [mean change score = −4 (−10 to 0) for 4 weeks (*p* < 0.001) and −6 (−10 to 0) for 12 weeks (*p* < 0.001)] than those in the control group [mean change score = −1 (−4 to 0) for 4 weeks (*p* = 0.006) and −3 (−9 to 0) for 12 weeks (*p* < 0.001)] (*p* < 0.001 for 4 weeks; *p* < 0.001 for 12 weeks). The changes in the CIPN sensory scores of patients in the control group [median change score = −1 (−7 to 0)] were significantly larger than those in the HGWD group [median change score = 0 (−3 to 0)] (*p* = 0.001) from the 4th week to the 12th week (Figure 2). Patients in the HGWD group also reported a larger decrease in CIPN motor score [mean change score = −2 (−5 to 0) for 4 weeks and 12 weeks (*p* < 0.001)] than those in the control group [mean change score = 0 (−2 to 0) for 4 weeks (*p* = 0.003) and 0 (−3 to 0) for 12 weeks (*p* = 0.004)] (*p* < 0.001 for 4 weeks; *p* < 0.001 for 12 weeks). The EORTC QLQ-CIPN20 autonomic scores showed no statistical significance between the two groups (*p* > 0.05) (Table 2).

### 3.3. Adverse Effects

The TRAEs (except neuropathy adverse events) are shown in Table 3. No grade 5 TRAEs were reported. All patients experienced grade 1 or grade 2 alopecia in the two groups. In the HGWD group and control group, the most common grade 3/4 TRAEs were neutropenia (6.5%) (*n* = 3), followed by ALT/AST elevation (2.2%). One patient experienced grade 3 leukopenia in the HGWD group. The most common grade 1 or 2 TRAEs were fatigue (43.5%), reduced appetite (30.4%) and ALT/AST elevation (19.6%) in the HGWD group, while there was fatigue (50.0%), reduced appetite (23.9%) and leucopenia (23.9%) and ALT/AST elevation (23.9%) in the control group. There was no significant difference in any grade TRAEs between the two groups (*p* > 0.05), with the exception of grade 1–2 leucopenia (*p* = 0.020). No patients withdrew from the study in the two groups because of TRAEs.

## 4. Discussion

Nab-PTX is one of the most important chemotherapeutic agents for the treatment of BC with a relatively high incidence of PN. Nab-PTX-induced PN is important dose-limiting toxicity that seriously affects the efficacy and use of Nab-PTX. Therefore, more attention should be paid to nab-PTX-induced PN during nab-PTX-containing treatment. Effective measures must be explored and discovered to prevent the development of nab-PTX-induced PN. To our knowledge, this study was the first prospective randomized controlled study to investigate the efficacy and safety of HGWD packs to treat nab-PTX-induced PN in patients with BC.

The pathogenesis of nab-PTX-induced PN has not been fully elucidated. Currently, it is believed that multiple factors are involved in its occurrence. The possible pathogenesis includes: (1) the chemotherapeutic agents act on the myelin pad and sensory cell bodies and axons of the dorsal root ganglia, which release the pro-inflammatory cytokines that activate apoptotic signaling cascades, alter central and peripheral neuronal excitability, and lead to epineurial shedding; (2) ion channel dysregulation; (3) microtubule disruption and axonal transport impairment; (4) mitochondrial dysfunction and oxidative stress; (5) trigger immune system multi-factor abnormalities (including abnormal cytokine secretion and abnormal immune cell function, etc.) leading to the development of neuroinflammation and sensitization of the sensory nervous system; (6) axonal degeneration; (7) injury to the sensory neurons of the dorsal root ganglion [2,18,19,20,21]. Although there are many methods for the prevention and treatment of PN, there is no specific effective medicine for the treatment of PN. Existing guidelines only recommended duloxetine as the first-line treatment for CIPN (including nab-PTX-induced PN) [22]. Other drugs for the treatment of CIPN included anti-epileptics, antidepressants, selective serotonin reuptake inhibitors, serotonin norepinephrine reuptake inhibitors, tricyclic antidepressants, and opioids, etc., all of which were derived from drugs used to treat other types of neuropathic pain. Some non-pharmacological therapies, such as exercise, scrambler therapy, cryotherapy, and compression therapy, were optional treatments for the management of CIPN [23,24,25]. However, existing treatments for CIPN have limited efficacy and still lack strong evidence for potential benefits.

HGWD exhibited effective, safe, less toxic, and fewer side-effects in the treatment of PN [13,14]. The effect and potential mechanisms of HGWD against paclitaxel-induced PN are unclear yet. Some underlying molecular mechanisms of HGWD in paclitaxel-induced treatment were as follows. A previous study showed that the standardized extract AC591 of HGWD played neuroprotective roles [26]. Another study showed that HGWD attenuated paclitaxel-related PN through inhibition of inflammation and oxidative stress via TLR4/NF-κB and PI3K/Akt-Nrf2 pathways [12]. The neuroprotective property of HGWD on paclitaxel-related PN provides important support to the potential application of HGWD for counteracting the side effects of paclitaxel during chemotherapy [12]. Some active chemical ingredients with their physicochemical properties screened from HGWD (15 compounds from Astragali Radix, 5 compounds from Cinnamomi Ramulus, 3 compounds from Paeonia Radix Alba, 11 compounds from Zingiberis Rhizoma Recens, and 7 compounds from Zingiberis Rhizoma) have been reported to possess anti-inflammatory and antioxidant activities, which indicated that these compounds may be key components for PN treatment [12]. HGWD also reduced inflammatory factors such as interleukin-4 (IL-4) and inducible nitric oxide synthase (iNOS) to perform anti-inflammatory effects [27]. A recent study has demonstrated that the mechanisms of action of HGWD in rats included relieving pain by raising the pain threshold [27]. A study also found that HGWD did not interfere with the antitumor activity of paclitaxel both in vitro and in vivo models [12]. These studies preliminarily demonstrated that HGWD could effectively attenuate paclitaxel-related PN without compromising the anti-tumor efficacy of paclitaxel.

To this date, the diagnostic criteria for nab-PTX-induced PN have not been established. The incidence and severity were often underestimated due to the under-reporting of patients and inadequate physician evaluation. One of the best assessment methods is to use an adverse drug reaction assessment scale, such as the EORTC QLQ-C30 scale that was used in this study. In the past, critical clinical management with the traditional Chinese herbal medicine of tingling/numbness in hands/feet was mainly given orally. However, patients often experienced gastrointestinal reactions, such as nausea and vomiting, and were reluctant to receive oral medication therapy after chemotherapy. Thus, the application of traditional Chinese herbal medicine immersing and washing limbs could alleviate the above adverse reactions [28,29,30]. Our study showed that almost all patients (97.8%) showed a decrease in CIPN20 sensory score and 40 patients (87.0%) showed a decrease in CIPN20 motor score after 12 weeks follow-up. There were significant differences in the sensory and motor score between the HGWD and control group (4 weeks: sensory *p* < 0.001, motor *p* < 0.001; 12 weeks: sensory *p* < 0.001, motor *p* < 0.001). The changes in the CIPN sensory scores of patients in the control group [mean change score = −1 (−4 to 0)] were significantly smaller than those in the HGWD group [mean change score = −4 (−10 to 0)] (*p* < 0.001) from the enrollment to the 4th week. However, the changes in the CIPN sensory scores of patients in the control group [median change score = −1 (−7 to 0)] were significantly larger than those in the HGWD group [median change score = 0 (−3 to 0)] (*p* = 0.001) from the 4th week to the 12th week (Figure 2). These results showed that patients immersed and washed limbs with HGWD packs after chemotherapy significantly alleviated patients’ peripheral neurotoxicity-induced symptoms, like tingling, numbness, etc., especially in the early phase of nab-PTX-induced PN.

Several previous systematic reviews and meta-analyses have not reported HGWD-related AEs [27,28]. This study also yielded similar results. HGWD was remarkably welltolerated in this BC population. No grade 5 adverse events were reported, and no patients withdrew from the study in two groups because of TRAEs. In the HGWD group and the control group, the most common grade 3/4 TRAEs were neutropenia (6.5%), followed by ALT/AST elevation. Only one patient experienced grade 3 leukopenia in the HGWD group. The most common grade 1 or 2 TRAEs were fatigue (43.5%), reduced appetite (30.4%) and ALT/AST elevation (19.6%) in the HGWD group. There was no significant difference in any grade TRAEs between the two groups (*p* > 0.05), with the exception of grade 1–2 leucopenia (*p* = 0.020). These results showed that HGWD packs are safe and tolerable in Chinese Patients with BC with nab-PTX-induced PN.

There remain some limitations to the present study. Firstly, the current grading standards for nab-PTX-induced PN are mainly based on the subjective feelings of patients. There is a lack of objective and reliable indicators for further verification. Secondly, the sample size of our current study is relatively small, and larger-scale studies are needed to further confirm the results of this study in the future.

## 5. Implications for Clinical Practice

With further in-depth experiments and clinical research, the potential pharmacological effects and value of HGWD packs are gradually being recognized. Our study found that immersing and washing limbs with HGWD packs could relieve patients’ nab-PTX-induced peripheral neurotoxic symptoms to some extent. However, there is still a lack of systematic research on HGWD packs. In-depth research on the pharmacokinetics and pharmacodynamics of HGWD will be a direction for future efforts, which will help to provide scientific theoretical data for expanding the scope of clinical application of HGWD. In clinical practice, when Western medications showed the limited effect on certain disease symptoms, we can also pay attention to the application of traditional Chinese herbal medicine for treatment.

## 6. Conclusions

Our study showed that there is a significant benefit in using HGWD packs to treat nab-PTX-induced PN. HGWD packs could significantly improve patients’ nab-PTX-induced peripheral neurotoxic symptoms, increase the tolerance for nab-PTX-containing chemotherapy, and further improve the quality of life of patients with BC. HGWD packs are also safe and tolerable in Chinese patients with BC with nab-PTX-induced PN.

## Figures and Tables

**Figure 1 jcm-12-00505-f001:**
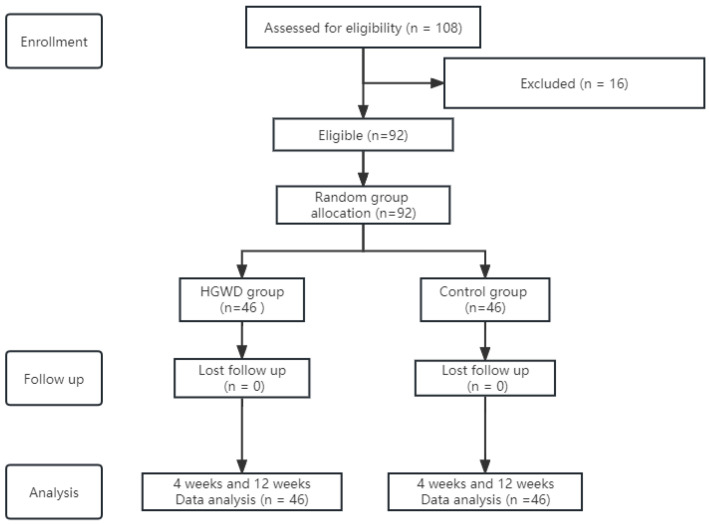
Patient selection of all patients in this study. Abbreviations: HGWD: Huangqi Guizhi Wuwu decoction.

**Figure 2 jcm-12-00505-f002:**
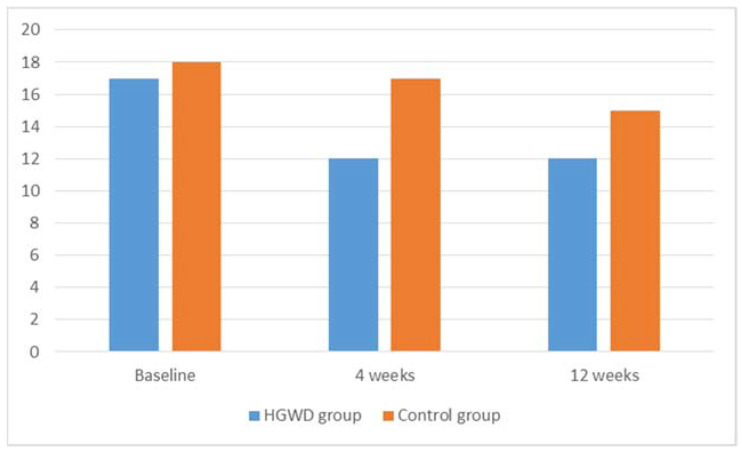
Median EORTC QLQ-CIPN20 sensory scores of HGWD group and control group. Abbreviation: EORTC QLQ-CIPN20: the European Organization for the Research and Treatment of Cancer-Chemotherapy-induced peripheral neuropathy 20; HGWD: Huangqi Guizhi Wuwu decoction.

**Table 1 jcm-12-00505-t001:** Baseline characteristics of eligible patients with breast cancer.

Variables	HGWD Group (*n* = 46)	Control Group (*n* = 46)	*p* *
Age (range)	50 (23–65)	52 (29–65)	
ECOG score			0.669
0	29 (63.0)	27 (58.7)	
1	17 (37.0)	19 (41.3)	
Menstruation status			0.381
Pre	27 (58.7)	28 (60.9)	
Post	19 (41.3)	18 (39.1)	
Lymph node metastasis			0.804
Yes	36 (78.3)	35 (76.1)	
No	10 (21.7)	11 (30.4)	
Pathologic tumor stage ^a^			0.620
I	8 (17.4)	6 (13.0)	
II	8 (17.4)	10 (21.7)	
III	23 (50.0)	19 (41.3)	
IV	7 (15.2)	11 (23.9)	
Molecular classification			0.513
Luminal A	3 (6.5)	5 (10.9)	
Luminal B	25 (54.3)	18 (39.1)	
HER-2	2 (4.3)	3 (6.5)	
TNBC	16 (34.8)	20 (43.5)	
Surgery			0.559
MRM	29 (63.0)	24 (52.2)	
BCS	11 (23.9)	15 (32.6)	
SLN	6 (13.0)	7 (15.2)	
Radiotherapy			0.369
Yes	38 (82.6)	41 (89.1)	
No	8 (17.4)	5 (10.9)	
Endocrine therapy			0.294
Yes	28 (60.9)	23 (50.0)	
No	18 (39.1)	23 (50.0)	
Anti-Her2 therapy			0.748
Yes	5 (10.9)	6 (13.0)	
No	41 (89.1)	40 (87.0)	

Note: ^a^ Staging was determined according to the 8th edition of the AJCC cancer staging. Manual; * Chi-square test. Abbreviation: HGWD: Huangqi Guizhi Wuwu decoction; ECOG, Eastern Cooperative Oncology Group; HER-2, human epithelial growth factor receptor 2; TNBC, triple-negative breast cancer; MRM, modified radical mastectomy; BCS, breast conservative surgery; SLN, simple mastectomy and sentinel lymph node biopsy.

**Table 2 jcm-12-00505-t002:** Median (range) EORTC QLQ-CIPN20 scores for primary outcomes of HGWD group and control group.

Outcomes	Median (Range) ^a^		Estimate of Median Change (Range, *p* Value) ^b^		Comparison of Change between Groups (*p* Value) ^c^
	HGWD Group	Control Group	HGWD Group	Control Group	
CIPN sensory					
Baseline	17 (12–24)	18 (12–24)	-		
4 weeks	12 (9–15)	17 (12–22)	−4 (−10 to 0, *p* < 0.001) *	−1 (−4 to 0, *p* = 0.006) *	*p* < 0.001 *
12 weeks	12 (9–15)	15 (11–19)	−6 (−10 to 0, *p* < 0.001) *	−3 (−9 to 0, *p* < 0.001) *	*p* < 0.001 *
CIPN motor					
Baseline	12 (10–15)	12 (10–14)	-		
4 weeks	10 (8–12)	12 (10–13)	−2 (−5 to 0, *p* < 0.001) *	0 (−2 to 0, *p* = 0.003) *	*p* < 0.001 *
12 weeks	10 (8–11)	12 (10–13)	−2 (−5 to 0, *p* < 0.001) *	0 (−3 to 0, *p* = 0.004) *	*p* < 0.001 *
Autonomic CIPN					
Baseline	2 (1–4)	2 (2–4)	-		
4 weeks	2 (1–3)	2 (1–3)	0 (−2 to 0, *p* = 0.157)	0 (−2 to 0, *p* = 0.102)	*p* = 0.633
12 weeks	2 (1–3)	2 (1–3)	0 (−2 to 0, *p* = 0.102)	0 (−2 to 0, *p* = 0.059)	*p* = 0.702

Note: * A p value < 0.05 was considered statistically significant; ^a^ Higher scores represent worse symptoms; ^b^ Wilcoxon matched-pairs rank test; ^c^ Mann-Whitney U tests. Abbreviation: EORTC QLQ-CIPN20: the European Organization for the Research and Treatment of Cancer-Chemotherapy-induced peripheral neuropathy 20; HGWD: Huangqi Guizhi Wuwu decoction; CIPN: chemotherapy-induced peripheral neuropathy.

**Table 3 jcm-12-00505-t003:** Adverse events related to treatment.

Adverse Events	Grade 1/2			Grade 3/4		*p*-Value
	No. (%)			No. (%)		
	HGWD Group (*N* = 46)	Control Group (*N* = 46)	*p*-Value	HGWD Group (*N* = 46)	Control Group (*N* = 46)	
Fatigue	20 (43.5)	23 (50.0)	0.531	0	0	-
Reduced appetite	14 (30.4)	11 (23.9)	0.482	0	0	-
Nausea/Vomiting	5 (10.9)	3 (6.5)	0.714	0	0	-
Constipation	2 (4.3)	2 (4.3)	1.000	0	0	-
Diarrhea	4 (8.7)	1 (2.2)	0.361	0	0	-
Skin rash	1 (2.2)	1 (2.2)	1.000	0	0	-
Oral ulcer	4 (8.7)	1 (2.2)	0.361	0	0	-
Alopecia	46 (100.0)	46 (100.0)	1.000	0	0	-
leucopenia	3 (6.5)	11 (23.9)	0.020 *	1 (2.2)	0	-
Neutropenia	7 (15.2)	8 (17.4)	0.778	3 (6.5)	3 (6.5)	1.000
Anemia	2 (4.3)	3 (6.5)	1.000	0	0	-
Thrombocytopenia	3 (6.5)	0	-	0	0	-
Creatinine elevation	1 (2.2)	4 (8.7)	0.361	0	0	-
ALT/AST elevation	9 (19.6)	11 (23.9)	0.613	1 (2.2)	1 (2.2)	1.000

Note: * A p value < 0.05 was considered statistically significant; Abbreviations: HGWD: Huangqi Guizhi Wuwu decoction; ALT: alanine transaminase; AST: aspartate transaminase.

## Data Availability

Data will be made available on request.

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
