# Peer review of "A Prospective, Randomized, Placebo-Controlled Study Assessing the Efficacy of Chinese Herbal Medicine (Huangqi Guizhi Wuwu Decoction) in the Treatment of Albumin-Bound Paclitaxel-Induced Peripheral Neuropathy"

_jcm, 2023, doi:10.3390/jcm12020505_

Round 1

Reviewer 1 Report

In excluded criteria you didn't mention the patient with peripheral circulatory (artery) insufficiention? did you exclude also these patients? please add some sentences about this 

Author Response

Point 1: In excluded criteria you didn't mention the patient with peripheral circulatory (artery) insufficiention? did you exclude also these patients? please add some sentences about this.

Response 1: Thank you for your valuable suggestion. Patients with peripheral vascular insufficiency were excluded in this study. We added it in the Method part: The exclusion criteria were:… (4) peripheral vascular insufficiency;…. (P2 Line 49-50).Your comments allowed us to significantly improve our manuscript. Thank you very much!

Reviewer 2 Report

Dear author

Thank you for the submission of your article to our journal. It is well known that taxan-chemotherapy plays an important role in the treatment of breast cancer both in the (Neo)adjuvant and metastatic settings. Each taxan has strong anti-tumor activity and causes various side effects. Concerning paclitaxel, peripheral neuropathy is the most important side effect. Many researchers have tackled with this side effect to date  but fave failed to treat or prevent this uncomfortable side effect. It is not clear from your paper which component of the herbal drug contributed to the prevention of peripheral neuropathy. Your paper, however, suggested that this herbal drug might have a small but preventive effect on texan-induced peripheral neuropathy.

Minor points

Adverse effects

Leukemia (23.9%), leukemia (P=0.020)

Are these misspelling of leucopenia?

P8

Line 6

Is the word “erotonin” a misspelling of serotonin?

Line 42-3 Closing parenthesis is missing.

Author Response

Point 1: Adverse effects:Leukemia (23.9%), leukemia (P=0.020), Are these misspelling of leucopenia?

Response 1: Thank you for your valuable suggestion. This is indeed a spelling error.We changed “leukemia” to “leukopenia” in the adverse effects part and discussion part.

Point 2: P8 Line 6 Is the word “erotonin” a misspelling of serotonin? Line 42-3 Closing parenthesis is missing.

Response 2: Thank you for this comment. We changed “erotonin” to “serotonin” (P8 Line 6). And we added closing parenthesis (P8 Line 42-3). Your comments allowed us to significantly improve our manuscript. Thank you very much!